# Hydrological Modelling and Climate Adaptation under Changing Climate: A Review with a Focus in Sub-Saharan Africa

Vincent Dzulani Banda [1,*], Rimuka Bloodless Dzwairo [2,3], Sudhir Kumar Singh [4] and Thokozani Kanyerere [1]

1   Department of Earth Sciences, University of the Western Cape, Private Bag X17, Bellville 7535, South Africa
2   Department of Civil Engineering, Durban University of Technology, Midlands, P.O. Box 101112, Imbali 3209, South Africa
3   Institute for Water and Wastewater Technology, Durban University of Technology, P.O. Box 1334, Durban 4001, South Africa
4   K. Banerjee Centre of Atmospheric and Ocean Studies, Nehru Science Centre, University of Allahabad, Prayagraj 211002, Uttar Pradesh, India
*   Correspondence: vsdbanda@gmail.com

**Abstract:** Empirical evidence continues to show that climate change remains a threat to the stability of the hydrologic system. As the climate system interacts with the hydrologic cycle, one significant repercussion of global warming includes changes in water availability at both regional and local scales. Climate change adaptation is intrinsically difficult to attain due to the dynamic earth system and lack of a comprehensive understanding of future climate and its associated uncertainties. Mostly in developing countries, climate adaptation is hampered by scarcity of good quality and adequate hydro-meteorological data. This article provides a synopsis of the modelling chain applied to investigate the response of the hydrologic system under changing climate, which includes choosing the appropriate global climate models, downscaling techniques, emission scenarios, and the approach to be used in hydrologic modelling. The conventional criteria for choosing a suitable hydrological model are discussed. The advancement of emission scenarios including the latest Shared Socioeconomic Pathways and their role in climate modelling, impact assessment, and adaptation, are also highlighted. This paper also discusses the uncertainties associated with modelling the hydrological impacts of climate change and the plausible approaches for reducing such uncertainties. Among the outcomes of this review include highlights of studies on the commonly used hydrological models for assessing the impact of climate change particularly in the sub-Saharan Africa region and some specific reviews in southern Africa. Further, the reviews show that as human systems keep on dominating within the earth system in several ways, effective modelling should involve coupling earth and human systems models as these may truly represent the bidirectional feedback experienced in the modern world. The paper concludes that adequate hydro-meteorological data is key to having a robust model and effective climate adaptation measures, hence in poorly gauged basins use of artificial neural networks and satellite datasets have shown to be successful tools, including for model calibration and validation.

**Keywords:** hydrological modelling; climate models; climate change adaptation; developing countries; emission scenarios; southern Africa

## 1. Introduction

Climate change has resulted in continued changes in the hydrologic systems in different watersheds across the globe [1,2]. Accordingly, hydrological research and assessment of climate change impacts is an existing matter at both global and regional levels [3,4]. In fact, climate change impacts are largely detected and well understood in natural systems, typically shifting precipitation and temperatures, which eventually change the hydrological system [5]. The common direct impacts of climate change on water resources include changes in the catchment water balance (increase or decrease) and nutrient cycling [6–8].

The hydrological system is largely influenced by among others climatic conditions, terrain characteristics, land use and land cover, and population growth [9], which collectively affect water availability and quality. The uncertainties in the magnitude and dynamics of these elements have resulted in challenges in the management and planning of water resources. Having reliable hydrological predictions is, therefore, important to developing climate-adaptive strategies for different water-related sectors [10–12]. This is particularly true at a regional or basin-scale because it is at these scales where the impacts are felt and adaptation strategies are developed and implemented [13].

Due to increasing concentrations of greenhouse gases and therefore global warming, the impact of climate change on water resources is expected to increase [14]. Achieving sustainable water management at a catchment scale will require predicting and analyzing future trends in water resources using advanced tools over long periods [15]. Further, assessing the impacts of climate change on water resources requires the application of different decisions on the method to be used. These include choosing suitable global climate models, downscaling techniques, emission scenarios, and also the approach to be used in hydrologic modelling [16]. This calls for a need for effective modelling approach in order to have rational water resource plans. For countries to successfully adapt to future changes in climate, there is a need for a comprehensive understanding of the climate situations in different regions in both seasonal, yearly, and decadal timescales. Thus, it is imperative to conduct research that will offer guidance and prediction on the extent of such impacts in order to devise proper adaptation responses. Certainly, the desirable approach moving into the future, is to try to identify the projected climate change impacts and devise strategies in anticipation [17].

This paper aims to provide a literature review of the applications, approaches, and the modelling chain involved in integrated hydrological modelling of climate change impacts. The paper focuses on key aspects involved in simulating future runoff at a catchment scale under changing climatic scenarios, for effective climate change adaptation. To accomplish the aforementioned aim, corresponding literature were obtained from water, hydrology, climate change, and climate/hydrological modelling journals. A literature search was performed using different search engines including Scopus, Google Scholar, ScienceDirect, and Web of Science. The search targeted internationally recognised peer-reviewed journals covering aspects of integrated climate and hydrological modelling. Materials obtained from the journals were complemented with information from reports mostly from the Intergovernmental Panel on Climate Change (IPCC).

To obtain the literature, different keywords and phrases were used, which included the following: 'climate change', 'climate adaptation and mitigation', 'hydrological processes', 'climate change impacts', 'climate models', 'hydrological modelling', 'model uncertainty', 'sub-Saharan Africa', 'southern Africa', 'developing nations', and 'climate emission scenarios'. Upon screening the titles and abstracts, only articles that focussed on climate change and hydrological models were considered. Considering the numerous available literature on the aforementioned themes, a "semi-systematic" approach was adopted for identifying and analysing the themes in literature since the aim was to have a narrative overview of the subject matter [18]. Although this process of article selection could be considered subjective and biased, it was adopted in this paper because it provided a broader and comprehensive context, current knowledge, gaps, and inconsistencies in the field under study. Global literature were used to provide a general understanding of integrated climate change and hydrological models and their associated modelling chain. Further, specific literature on climate change and hydrological modelling from sub-Saharan and southern Africa for the past decade (2012 to 2022), were randomly selected due to the regions' recognized high vulnerability to climate change. Consequently, this paper did not discuss every single available literature relating to the subject at hand, instead, only representative studies (over 200 articles) were considered for the qualitative synthesis based on the above mentioned themes.

First, the article introduces the challenges associated with climate change adaptation in developing countries particularly in sub-Saharan Africa in Section 1. In Section 2, the paper highlights the relationship between climate change, and anthropogenic activities and how they influence the hydrological processes. Sections 4 and 5 provide reviews on global and regional climate models and climate change scenarios for integration into hydrological models. In Section 6, a discussion of the southern African region is provided focusing on a literature review of some of the applied global and regional climate models for use in hydrological modelling. This is followed by a review of the common hydrological models used in climate change analysis and their associated uncertainties and opportunities for reducing the uncertainties in Sections 7 and 8. Finally, conclusions derived from the review and potential future research are given in Section 9.

## 2. Climate Change Mitigation and Adaptation in Sub-Saharan Africa

Sub-Saharan Africa is the world's most vulnerable region to climate change due to low adaptive capacity resulting from development challenges at different levels and sectors [19–21]. The intensity and frequency of climate-related risks in this region are further aggravated by rising temperatures and sea levels as well as rainfall variability [22]. These include, among others, the increase in frequency and duration of dry days in Zambia [23], the persistent drought observed between 2015 and 2020 in South Africa [24,25], flooding due to tropical cyclones in Malawi, Zimbabwe and Mozambique observed recently in 2019, 2021 and 2022 [26–28] and the severe rainfall observed in the Eastern Africa region between October 2019 and January 2020 [29]. Meanwhile, risks to climate change are largely managed through two primary approaches, namely, mitigation, and adaptation [30]. Mitigation seeks to lessen or control the magnitude of climate change impacts by reducing the human factors contributing to climate change, particularly greenhouse gases. Adaptation emphasizes reducing climate change damages and capitalizing on the opportunities associated with it, for instance, by forecasting its trends and implications [31].

Mitigation is mainly considered to be a concerning matter for developed countries while adaptation is understood to be a most important aspect for developing nations [32–35]. Adaptation to climate change has different overarching goals including increasing effective response to stresses and managing risks, increasing resilience and continued functioning of a system, and reducing vulnerability during times of hazards [36]. Hence, to achieve such goals it is vital to proactively implement the different adaptation measures, which is also key for enhancing the socio-economic development of countries [22,37].

In the context of expected future climatic variability, countries in sub-Saharan Africa are investing in different adaptation strategies. For instance, the southern African region is implementing programs to promote the conjunctive use of surface water and groundwater and other augmentation schemes like desalination, to reduce the total reliance on surface water, particularly in urban areas [38,39]. In the coastal town of Knysna in South Africa, the municipality is engaged in the maintenance of sea walls to protect the infrastructure from sea level rises [40]. The use of indigenous knowledge for weather forecasting and water conservation measures is one of the common measures applied in sub-Saharan African countries [41–44]. For example, to cope with changing precipitation and heat stress, some local communities in Zimbabwe harvest water by digging wells for use during water scarcity [45]. Meanwhile in Senegal, in addition to implementing climate change awareness campaigns, the country is increasing water supply by constructing dams and boreholes, which are supported by the planning instruments such as the National Adaptation Plans [46]. However, it has been revealed that most of these adaptation measures are put in place after the occurrence of a crisis (e.g., drought or flood) [47,48], hence the measures are based on historical weather data [49] and the current adaptation gaps. This may not be a viable and reasonable approach because the adaptation measures need to be sustainable by incorporating both current and future projected climate change including extreme events. Overall, it is not enough to only adapt to the existing adaptation deficit and there is a need for long-term adaptation measures based on the climate projections.

Furthermore, it has been shown that climate adaptation strategies in sub-Saharan Africa are dominated and led by the local communities and individuals [43,50–52]. Thus, locally relevant climate information and projections suiting the environmental conditions at that local scale cannot be overemphasized.

Essentially, climate adaptation is likely to be effective with good quality and adequate hydro-meteorological and environmental data. For instance, according to the Global Center on Adaptation (GCA) report based on studies in Africa [53], the possible benefit to cost ratios attained from early adaptation is generally high throughout most climate adaptation measures, as shown in Figure 1. From the GCA review, it was established that better knowledge and availability of weather and climate information services is more vital in achieving climate adaptation than any other adaptation strategies [53], suggesting the need to intensify such initiatives in so far as climate adaptation is concerned.

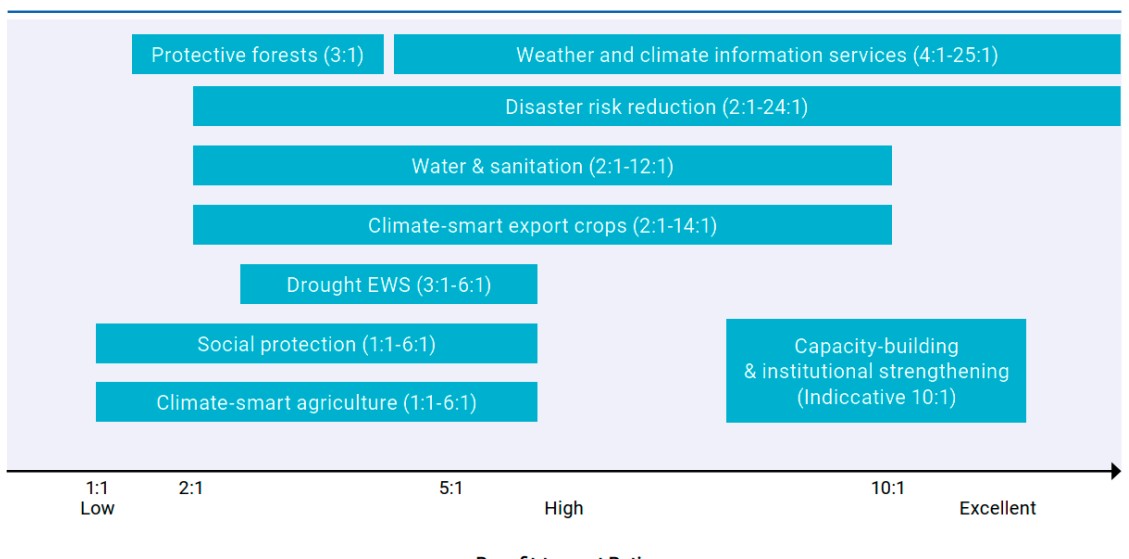

**Figure 1.** The representative benefit-to-cost ratio ranges for different adaptation measures based on information review from Africa. Photo credit: The Global Center on Adaptation [53].

Climate adaptation planning is hampered by a lack of reliable information on climate including the deep uncertainties relating to the timing and spatial distribution of the impacts [37]. Thus, it remains a challenge to thoroughly understand the extent of future vulnerability to both human and natural systems including water resources. Nevertheless, hydrologists are required to provide information on the future climate and water resource trends to water resource managers and policy-makers for future development and management. Hydrological and climate models are, therefore, used to predict the effects of climate change on water resources and to forecast the potential hydrological regimes in the future [54]. Predicting and quantifying the hydrological impacts and extent of future climate change is a necessity for devising climate adaptive measures. Prediction of the impacts of climate change on the hydrological systems including streamflow is usually performed by using a series of modelling approaches. At first, future changes in the atmospheric energy or the representative concentration pathways are identified, followed by the compilation of climate projections generated by the general circulation and regional climate models, and finally, the impact assessment is achieved by applying the chosen hydrological impact models under different radiative forcing and climate scenarios [55]. This entails the need to adequately understand the modelling chain for improved prediction and effective adaptation of climate change impacts.

### 3. Nexus between Climate Change, Land Use, and Hydrological Processes

Hydrological processes within a river basin are largely affected by two key factors namely, changes in climate elements (e.g., precipitation and potential evapotranspiration) and changes in catchment land use [17,56]. Changes in land use and anthropogenic climate changes are facilitated by population growth, urbanization, and the need to develop better facilities [57,58]. Basically, knowledge of the collective impacts of climate change, socio-economic development, and associated land-use change on water resources is critical for designing long-term water resource planning. Indeed, with the certainty of climate change and a changing natural environment, the availability of water resources in the future is heavily reliant on proper land use planning and management [59]. Climate and land-use changes can result in both increase or decrease in streamflow, hence there is a possibility that the changes due to the two factors may counterbalance [60]. Changes in climate and catchment surface characteristics will result in the spatial and temporal heterogeneity of water resources. It is thus important to understand such combined impacts for accurate prediction of future changes in hydrological processes.

Due to the complexity of the hydrologic system owing to the variability in land use and climate, it is becoming increasingly important to create and integrate different data sets and approaches to better understand the system [61]. Hydrological models based on the land surface data coupled with regional climate models are therefore used for forecasting the behaviour of hydrological and meteorological events under different climate scenarios. These climate scenarios are noticeable tools that are useful to decision-makers with respect to characterizing the future climate. Sutton [62] argued that reasonable modelling and prediction of future hydrological events will need to incorporate socio-economic data including future population projections. This is particularly true as catchment land use characteristics are largely modified by population changes and the eventual economic activities that have an influence on the hydrological processes.

To date, there have been numerous studies that have been documented and provided reviews about the understanding of the combined impacts of climate and land use changes on the hydrological processes including streamflow quantity [63–67]. For better planning of future water quantity within a river basin, hydrological simulations should incorporate both the projected climate change and future land use [68]. Being the major determining factors influencing the hydrological regimes, it is vital to differentiate the distinct impacts induced by climatic and land use changes for appropriate climate change adaptation and land use planning [69,70], especially due to the ever-changing climate, population dynamics, and associated land use patterns.

Several approaches have been used in literature to separate and understand the intricate and interconnected relationship between climate and land use changes with the hydrological regime. Some studies have used only hydrological models, e.g., [71], others have used a combination of conventional statistical methods such as the Mann–Kendall and regression analysis with hydrological models, e.g., [72,73], other studies employed a paired catchment approach, e.g., [74] while some have used conceptual approaches like climate elasticity and sensitivity methods [75]. Meanwhile, Garg, Nikam, Thakur, Aggarwal, Gupta and Srivastav [59] argued that the best approach would be the physically based and spatially distributed hydrological models because of their potential to simulate hydrological processes in different dimensions. Linear regression is relatively convenient and easy to use, but it does not consider the non-linearity experienced in the hydrologic system [76]. The paired catchment approach is challenging as it requires one to identify a homogeneous catchment with matching climatic characteristics to the one being studied [77]. The use of hydrological models is encouraged because models are able to show the physical mechanism and reflect the changes in streamflow [78], however, the approach requires high quantities and good quality data leading to numerous uncertainties. Therefore, most researchers use a mix of approaches due to limitations and assumptions associated with each individual method, because the different methods might produce contrasting results. For example, Guan, et al. [79] noted in their study that the conceptual

Budyko-based decomposition showed streamflow variation being largely attributed to physical changes in catchment characteristics while the hydrologic model method indicated climate change to have a major influence on streamflow changes.

To better understand the application of the above-mentioned methods, land use scenarios need to be well examined, including the past and projected or future land use. Further, when examining the impact of land-use changes on the hydrological regime, it is imperative to choose a desirable hydrological model that can definitively handle the spatial and temporal changes in the river basin's physical characteristics [80]. Hence, the chosen land-use model should be capable to generate realistic land-use projections because this will have an overall impact on the hydrological model output. The need for accurate land-use projections is necessitated by the fact that in some catchments where the effects of climate change are not felt, increasing population will always create a burden on water resources due to land use/cover associated changes such as evapotranspiration, runoff, and groundwater recharge [69].

## 4. Climate Change Modelling

General Circulation Models (GCMs) are essential and currently have become potential tools for simulating global climate change and variability [81,82]. These numerical coupled models represent various earth systems including the atmosphere, oceans, land surface, and sea-ice [83], and are capable of providing long-term trends in historical, present, and future climate. However, GCMs fail to provide detailed impacts of climate change on hydrological processes in small and mid-sized catchments at regional and local scales [84], and these models only provide rough estimations of uncertainties [85]. The GCMs provide climate projections at an exceptionally coarser scale of around 250 km × 250 km [86,87]. However, recent developments such as the High-Resolution Model Intercomparison Project (HighResMIP) within the CMIP6 provide an important horizontal resolution at a scale finer than 50 km [88], thereby providing reliable simulations of the physical processes [89]. Nevertheless, GCMs may not be sufficient to provide detailed regional information for climate change impact studies including hydrological studies at river basin scale.

Most climate change impact, adaptation, and vulnerability studies are focused on a regional or local level, hence climate change information needs to be available at a finer spatial resolution [90]. Accordingly, GCM outputs are integrated with regional models for better hydrological modelling at that local scale. Ye and Grimm [84] pointed out that downscaled GCM outputs are incorporated as input to physical process-based models that predict hydrologic trends and the potential impacts of climate changes at a catchment scale. Sun, et al. [91] assented that in general, the common approach for analyzing climate change impacts on hydrology is by pairing climate variables from GCMs with hydrologic models. Moreover, assessing the hydrological impacts of climate change comprises projecting the climate at a global scale using the GCMs, downscaling the global projections to a regional scale using regional climate models and/or statistical models, and finally, using the regional outputs in the hydrological modelling [92]. Although global hydrological models are also used to simulate global river flows [93–95], regional models have the potential to provide solutions regarding the hydrologic variability at a regional scale, which is primarily of interest to hydrologists. This is specifically true because despite the fact that climate change impacts are perceived to be experienced at a global scale, the primary interest is how society responds to the local and regional effects that have been facilitated by global deviations [96].

Regional Climate Models (RCMs) serves to improve the spatial and temporal resolution of the coarser GCMs, thereby providing better data for informing policy. The rationale for the formulation of RCMs and their significance led to the notion of "downscaling", whose purpose is to refine different data irrespective of its resolution [97,98]. Thus, downscaling can simply be described as a process whereby local and regional climate information is derived from large-scale modelled data. Downscaling works on the principle that the GCM describes the effect of large-scale climate forcings, such as due to changes in solar radiation and influx of greenhouse gases, whereas the RCMs improve the resolution of this

large-scale data by incorporating the smaller components of the GCM grid scale forcings like topographical and land cover variations at a mesoscale level [83,90].

Under downscaling, there are two main approaches to how information on the local conditions can be combined with large-scale climate projections, namely statistical and dynamical downscaling. Statistical downscaling entails developing an empirical relationship between large-scale climate variables that are provided by the GCMs and the fine resolution observed from the local climate, which is then used to simulate the future climate [99,100]. It is based on the concept that climate at a regional scale is largely influenced by climate experienced at a larger scale and the existing local features, thus, large-scale climatic parameters serve as predictors for local and regional parameters [98]. In contrast, dynamical downscaling involves using a GCM output as an input for RCM. In essence, RCM uses large-scale climate data provided by the GCM at the lateral boundaries, which are combined with the local scale features like land–sea contrast, topography, and land-use variations [101], thereby producing representative climate information at a resolution range of 2.5 to 100 km$^2$ [102]. The statistical downscaling approach is preferred because, apart from requiring less expensive computing resources [83], it is a viable method for developing precise and local-level climate estimates [86]. However, training the statistical models requires long-term and higher spatial resolution climatic data [99], which is a common challenge in developing countries.

To date, RCMs have been used in different applications that include: providing input data for assessing the impact of climate change, investigating climate variability, projecting future climates, and largely downscaling the hydrological cycle [103]. Despite the additional value in RCMs, as earlier noted, their resolution may not appropriately represent the local catchment characteristics due to bias errors between the climate model and the observed data [104–106]. Therefore, before performing any climate change impact studies, there is still a need to adjust the regional climate model simulations to match the spatial resolution required by the impact hydrological models, through what is referred to as bias correction [107–109]. Generally, bias correction is performed through the use of distributional model output statistics such that the simulated and observed probability distributions are within the same ranges [110,111]. To rescale these RCM outputs, a number of statistical bias correction approaches have been developed and available in the literature [112–114]. The most prevalent ones are the univariate bias correction methods whereby only one RCM simulated variable is corrected at a particular time and location [115,116]. Although these univariate methods are considered simple and thus their prevalence, they may be unable to reestablish the spatial and temporal relationships between the climatic variables thereby creating more biases [117,118]. Consequently, multi-variate bias correction techniques, that ensure that there is uniformity in spatiotemporal fields and multiple climatic variables such as precipitation and temperature, have been developed. Multi-variate bias corrections are considered to result in improved estimation of the joint likelihood of precipitation and temperature [119]. To date, researchers [115,117,120] are evaluating the applicability and effectiveness of using the multivariate bias correction methods taking into consideration the different assumptions in the method. For instance, a study by Guo, Chen, Zhang, Xu and Chen [120] showed that the use of multivariate methods improves the simulation of hydrological variables including evaporation and streamflow.

Generally, choosing the right bias correction method is required because these methods can also be a source of uncertainty in the modelling process, especially when simulating streamflow [121,122]. In a nutshell, the assessment of climate change impacts on hydrology is performed in three key stages. Firstly, future climate projections are generated using GCMs and RCMs, then the climate model outputs are used for hydrological modelling. Finally, the hydrological model findings are statistically analyzed so that they can be used for decision-making in different sectors.

## 5. Assessing Future Climate Change: Climate Change Scenarios

As earlier indicated, water resource managers and other related authorities are required to make decisions on how best to plan and manage water resources based on the expected climate for a particular region. Yet, quantifying the radiative forcing based on the concentration of future greenhouse gases is a challenge. This is because the concentration of greenhouse gases is dependent on varying driving forces, among others, advancement in technologies, population growth, energy sources, political will, and changes in attitudes towards the environment [123]. Hence, it is a best practice for climate change practitioners to use a variety of future climate change scenarios that take varying levels of greenhouse gas emissions into account, rather than a single set of future climate data. This entails that professionals in the climate sector can work with future climate data to investigate potential climate conditions over a wide range of futures. These futures are defined through emission scenarios, which are used to evaluate the effects of near-term decisions on the long-term future [124]. In general, emission scenarios can be described as "images of the future" as they suggest what the future will look like [125], thus they are useful tools for analyzing how the various driving forces (human activities) may influence future emissions [126]. Accordingly, emission scenarios help in analyzing climate change, which includes climate modelling, impact assessment, adaptation, and mitigation [125].

In the past decades, climate scientists have developed a scenario framework that integrates a variety of potential climate futures and society for better policy-making. Recently, a framework consisting of potential atmospheric and socioeconomic pathways has been developed and adopted by the climate research community, to what is referred to as the Shared Socioeconomic Pathway (SSP)–Representative Concentration Pathway (RCP) framework [127]. This framework provides diverse potential future information such as climate, society, and economic trends, thereby offering an essential platform for climate research. The RCP and SSP scenarios are discussed in this section.

To generate future climate scenarios due to changing information needs for policy-makers, the IPCC Fifth Assessment Report [5] published a standardized set of scenarios called Representative Concentration Pathways (RCPs) that are used for climate research and as input for running climate models. RCPs are a collection of scenarios that encompass the impact of emissions, concentrations, land use/land cover, and various climate policies [128]. These are different from the scenarios from the Special Report on Emission Scenarios (SRES) [125], which only considered the radiative forcing as a result of greenhouse gases and aerosols from anthropogenic factors without any mitigation policies to limit greenhouse gas emissions.

These four RCP scenarios are: (i) RCP2.6, a low greenhouse gas (GHG) emissions scenario largely characterized by the declining use of oil and low energy intensity (ii) RCP4.5 describing medium-range emission with low energy intensity and strong reforestation programs (iii) RCP 6.0 describing medium-range emission and is consistent with heavy reliance on fossil fuels and intermediate energy intensity (iv) RCP 8.0 representing very high GHG emissions due to no policy changes to reduce emissions, whereby by 2100 the global temperatures will have increased by 2.6 °C and 4.8 °C than it was in 2000 [5].

On the contrary, SSPs are the latest scenarios used for the IPCC Sixth Assessment Report (AR6) and Coupled Model Intercomparison Project Phase 6 CMIP6 [129]. SSPs provide key narratives for different possible changes in the world climate in the future (over the century) when there is no climate policy. They comprise two elements, namely: narrative storylines and quantitative measurements of potential changes in socioeconomic indicators such as demographics, economic development, and technological advancement [126]. The development of SSPs was facilitated by the need to advance scenarios that cover both the greenhouse gas concentrations and the role of social-economic factors in global warming, hence they complement the RCPs. Further, the development of new scenarios may always be required to incorporate and run the latest generation of climate models.

Ultimately, the SSPs have variables from six main categories [130]: economy and lifestyle, demography, technology, institutions and policies (but not climate policies), envi-

ronment, and natural resources. Five SSPs have been developed, and their outcomes reflect the possible combination of challenges the society might face while implementing the mitigation and adaptation strategies. An overview of the five SSPs storylines is presented in Table 1.

**Table 1.** A synopsis of the SSP and their narratives. Adapted from [130].

| Shared Socioeconomic Pathway | A Summary of the Narrative |
|---|---|
| SSP1: Sustainability—Taking the green road (low challenges to mitigation and adaptation) | Its emphasis is on the commitment to achieving development goals, increasing environmental awareness in societies around the world, and a gradual move toward less resource-intensive lifestyles. A society following the SSP1 path will experience relatively low challenges to mitigation due to the increase in renewable energy and environmentally friendly technologies. Again, relatively low challenges to adaptation will be experienced because of the reductions in inequality and strong institutions at global to national scales. |
| SSP2: Middle of the road—(intermediate challenges to mitigation and adaptation) | Describes a scenario whereby the society follows a path in which the social, economic, and technological trends do not significantly change from the historical patterns. Institutions at the national and international levels strive to but move slowly towards accomplishing sustainable development goals. Environmental degradation, global population growth, and education investments will all be moderate. Hence, the world will experience moderate challenges to mitigation and adaptation, but with major differences amongst countries. |
| SSP3: Regional rivalry—A rocky road (high challenges to mitigation and adaptation) | This pathway is characterized by increasing nationalism, concerns about competitiveness and security, and regional conflicts, which eventually push countries to highly focus on domestic and regional issues. Further, economic growth is slow, inequalities are high while population growth is high in developing countries and low in developed nations. |
| SSP4: Inequality—A road divided (low mitigation challenges, adaptation challenges dominate) | Inequalities increase due to increasingly unequal investments in human capital, economic opportunity, and political power. Economic growth is moderate in industrialized and middle-income countries, while low-income countries remain behind. Low challenges to mitigation are largely due to the improved investment and adoption of low carbon energy sources while adaptation challenges will be high for societies with low levels of development and little access to effective institutions for coping with economic or environmental stresses. |
| SSP5: Fossil-fueled development—Taking the highway (high challenges to mitigation, low challenges to adaptation) | Society highly relies on competitive markets, innovation, and participatory societies to produce rapid technological progress and the development of human capital as a path to sustainable development. Global markets are increasingly integrated and there are strong investments in health, education, and institutions to enhance human and social capital. Further, the push for economic and social development is coupled with the exploitation of abundant fossil fuel resources and the adoption of resource and energy-intensive lifestyles around the world. High challenges to mitigation are a result of heavy reliance on fossil fuels and the lack of global environmental concern while relatively low challenges to adaptation are due to the attainment of human development goals, robust economic growth, and highly engineered infrastructure. |

It can be debated that on their own, SSPs are as well limited as they produce climate projections that are deemed not to specifically relate to a societal pathway. Hence, due to the limitation of RCPs, it is recommended that RCPs and SSPs are merged for a better assessment of climate risks and mitigation and adaptation policies [127].

The use of a combined RCP/SSP framework in hydrological modelling is useful in having a comprehensive knowledge of future climate impact on river basins under various climate emission scenarios. Accordingly, researchers in the hydrology and climate field employ the RCP/SSP scenarios to better understand the potential future changes in the water resources under different climate scenarios. For instance, Kumar, et al. [131] employed the Soil and Water Assessment Tool (SWAT) model and used climate projections from CMIP5 and CMIP6 to understand the dynamics of hydrological and environmental flows in the central Himalayan River basin (India). In the study, future climate projections for rainfall and temperature were evaluated using both the RCPs and SSPs emission

scenarios. Mensah, et al. [132] used the WEAP model together with two RCPs (RCP4.5 and RCP 8.5) and three SSPs (SPPS 2, 3, and 5) to simulate the groundwater demand at present and future times in nine sub-catchments of the White Volta River Basin in Ghana. The study established that four of the nine sub-basins will face water shortage under all future scenarios. The study further showed that climate change and an increase in socioeconomic activities will create gaps between groundwater demand and supply, particularly in areas with higher population density and arable agricultural land [132].

Meanwhile, since the advent of the CMIP6, other researchers in recent years have been using the CMPI6 model only for climate projections because the new and updated model has higher climate sensitivity and mostly produces simulated data closer to the actual observed climate change because it incorporated experiences from the past models and used new technologies [133,134]. For instance, Song, et al. [135] used eleven CMIP6 GCMs to estimate future runoff for two SSPs using the SWAT and Long Short-Term Memory networks (LSTM). Coupled with the SWAT model, Zhang, et al. [136] employed data from the CMIP6 program to simulate the runoff in the Baihe River Basin (China) based on three shared socioeconomic pathways (SSP1, 2, and 5).

Studies that have applied the CMIP6 in southern Africa are not prevalent, however, a few researchers have employed the CMIP6 data to project the climatic parameters including precipitation and temperature, mostly at a regional level. For example, Almazroui, et al. [137] applied three SSPs scenarios (SSP1, 2, 5) from CMIP6 to project changes in precipitation and temperature over Africa. The study observed significant increasing trends in temperature and high precipitation decline in southern Africa under all three scenarios. Sian, et al. [138] effectively used data from the CMIP6 to model precipitation trends over the southern Africa region, however, there were discrepancies in the results in the different sub-regions. Therefore, it is recommended that the applicability of the CMIP6 program be tested in different countries and localized river basins in southern Africa to demonstrate its performance and relevance in predicting future climatic variables such as temperature, precipitation, and runoff in the region. This is notably the case for the majority of southern Africa because the response and adaptation to climate change effects are generally achieved from the analysis of local conditions [139].

From the reviews, it is highlighted that emission scenarios are not considered to be climate predictions, largely because there is high uncertainty in the future. Therefore, it is important to explore various emission scenarios and regularly reassess these scenarios to reflect technological and socioeconomic changes [140]. Applying multi-model ensemble scenarios, including the recent AR5/AR6 (RCPs/SSPs) into various climate and hydrological models has the capability of increasing the projection range that may be closer to reality. The combination of RCPs and SSPs employed in the CMIP6 model ensured that the potential future socioeconomic changes are better captured and described, which is key to having more rational future scenarios.

## 6. Synopsis of Applied GCMs and RCMs in Southern Africa

Currently, climate change modelling studies at a regional and local scale entail starting with a global climate model and then downscaling it to the desired region. Thus, the chosen GCM and RCM will not only have a direct implication on the regional or local water resources but also, in general, climate change adaptation. The key challenge in choosing the global models for use in regional studies relates to the fact that different GCMs produce varying downscaled results [141,142]. Therefore, for appropriate water resource planning, selecting a suitable GCM for a specific location is a necessity.

As the case for the majority of Africa, the southern African region is considered to be one of the regions highly vulnerable to the effects of climate change and variability [138], largely due to its low adaptive capacity [20] and because a majority of its people directly depend on the natural environment for a living [143]. To formulate effective climate adaptation approaches, scientific information on the possible future climate is needed for every region including at a local scale. To this effect, the demand for climate change

information at different spatial and temporal resolutions is gaining popularity amongst a wide spectrum of decision-makers. Hence, climate models are the major mechanism for projecting future climate change under different emission scenarios [144].

Numerous researchers have to date assessed the performance of climate models to simulate the different climate data over the southern African region. The studies focused on projecting information such as surface solar radiation, droughts, precipitation variability, climate change signals, and hydrological features such as runoff and evapotranspiration [145]. At present, the Coordinated Regional Downscaling Experiment (CORDEX) Africa domain is the most common RCM that has been evaluated to establish its ability to simulate climate data including in southern Africa.

Shongwe, et al. [146] performed a study to evaluate the capability of CORDEX RCM in simulating variation, timing, and frequency of events in monthly summer rainfall over southern Africa. The study observed that majority of CORDEX RCMs were able to simulate the progression, extent, start, and cessation of precipitation over southern Africa, despite the presence of biases in some models. The observed and modelled data strongly correlated around the Inter-Tropical Convergence Zone, near the 20° South region. Shongwe, Lennard, Liebmann, Kalognomou, Ntsangwane and Pinto [146] further noted that amongst the individual RCMs, the ARPEGE5.1 showed to have performed better within the southern Africa region.

Meque and Abiodun [147] assessed the ability of ten RCMs from the CORDEX project to simulate the relationship between El Nino Southern Oscillations and the droughts in southern Africa. Of the ten RCMs, i.e., RegCM3, ARPEGE, PRECIS, CRCM, HIRHAM, REMO, RACMO, RCA, CCLM, and WRF, the ARPEGE model performed the best while the worst simulation was from the CRCM model. The disparities in the model performance were attributed to the lateral boundary conditions of the models.

Using data from 23 GCMs from the CMIP6, Sian, Wang, Ayugi, Nooni and Ongoma [138] explored the spatial-temporal changes and modelled the future trends in precipitation over southern Africa. The study established that the GCMs were effective in capturing the precipitation trends over the region, however, precipitation was generally overestimated in high elevation areas. Further, the study noted that three GCMs namely, FGOALS-g3, MPI-ESM1-2-HR, and NorESM2-LM, performed the best over the region. Based on the findings from the southern Africa sub-regions (i.e., Eastern-Southern Africa, Western-Southern Africa, and Madagascar), Sian, Wang, Ayugi, Nooni and Ongoma [138] concluded that there is no single GCM that produced consistent results, hence, precipitation trends cannot be generalized for the region and precise findings can be obtained through small scale local studies.

A study by Karypidou, et al. [148] was conducted to see if the climate change signals and monthly precipitation bias and variability within the southern Africa region were a result of RCMs or their driving GCMs. An ensemble of 19 RCMs from the CORDEX-Africa namely, CCLM4-8-17.v1, RCA4.v1, and REMO2009.v1 and a collection of 10 GCMs from CMPI5 were used. The study noted that in the simulations there was a consistent wet bias in both RCM and GCM, although the magnitude and spatial extent were smaller in the RCMs. Thus, the study concluded that RCMs could be viable tools for studying the impacts of climate change in the southern African region as they can resolve some biases observed in GCMs. Additionally, Karypidou, Sobolowski, Katragkou, Sangelantoni and Nikulin [148] showed that projections from CORDEX-Africa domain can provide sound information for studying climate change impacts. Using CORDEX-Africa, CMIP5, and CMIP6 ensembles to determine precipitation trends in southern Africa, the study showed that CORDEX and CMIP5 data underestimated the observed trends while the CMIP6 showed conflicting results by indicating a continuous drying trend. Nevertheless, a better performance was observed in the CORDEX-Africa, particularly on the annual and extreme precipitation indices observations.

Based on the aforementioned recent studies, it is evident that research on climate modelling within the southern African region is expanding. Moving forward, it is critical

to understand how well these models represent a specific country at a local scale, which is a crucial element for carrying out climate change impact assessments and adaptations for a specific context. This is particularly important because, for instance, within Africa biases in RCMs acquired from GCMs vary with regions and climatic variables, hence specific impact assessment for a particular country or local region needs its model evaluation separately [149]. Further, characterizing the strengths, weaknesses, and uncertainties associated with the climate models is a prerequisite for projecting future climate change impacts and model evaluation. In some studies, for example [150], it has been seen that there is a poor correlation between the individual RCMs and GCMs with their corresponding observed data. Therefore, a collection of RCMs should be used based on the RCM's capability to model climatic features of the specific region under interest. Again, the differences in the models in projecting the climatic variables can be minimized by reducing the biases from both the GCM and RCM, hence enhancing the reliability of the data [151].

## 7. Hydrological Models under Changing Climate: Role of Hydrological Models in Climate Change Studies

There exists an intricate relationship between hydrological processes such as stream-flow and climatic variables like temperature and rainfall across many river basins. As such, to examine the effects of climate change on streamflow, a mix of hydrological models and climatic projections from GCM and RCMs are commonly utilized [81,152]. Despite the potential to provide data in a relatively finer resolution, RCMs solely may fail to furnish the appropriate aspects for addressing the climate change on water resources and hydrological processes. In general, there is high uncertainty in future projections of water resources due to the disparities in the high-resolution climate models and the hydrological models at a local basin scale [153]. In contrast, Olsson, Arheimer, Borris, Donnelly, Foster, Nikulin, Persson, Perttu, Uvo, Viklander and Yang [2] argued that the matching in sizes and spatial scales (less than 50 km) between most river basins and a general RCM, would mean that the RCMs, despite the biases, can provide pertinent spatial-temporal climate simulations. Regardless, RCM outputs are employed as input data and are integrated with the hydrological model to assess basin-scale hydrological processes [145,154]. This is relevant to the extent that some hydrological impacts are assessed at a highly local scale having a temporal climate variability, which has fine resolutions compared to those offered by a standard RCM.

To date, hydrological models are widely regarded as essential tools for managing environmental and water resources. Thus, the modelling of climate change impacts on hydrological processes continues to receive interest from the different land surface and hydrology researchers, considering the continued land use/cover changes and varying climate characteristics. As previously highlighted, the impact assessment of climate change on hydrological processes encompasses many stages that include choosing the GCM, the downscaling method, the emission scenario and also deciding the approach to hydrologic modelling. For instance, Ref. [16] noted that selecting the structure of the model and appropriate calibration of parameters are two of the most important elements to consider. When these two elements of the hydrologic model are chosen based on the modeller's preferences, the impacts of climate change are perceived differently by different modellers, hence having an impact on the climate adaptative strategies [16]. This section reviews some of the common criteria used when selecting a hydrologic model for climate impact assessment with a comparison between studies in developed and developing regions with a focus on southern Africa.

### 7.1. Choosing the Right Hydrologic Model for Climate Impact Analysis

Hydrological models are widely employed to predict future stream runoff as a way to analyze climate change impacts on water resources. Several studies on the climate change impacts on hydrological processes have shown that, for instance, in southern Africa, there is a high likelihood of temperature increase in the future while precipitation and streamflow

could increase or decrease depending on the region, the extent of catchment modification, and mostly the selected GCM/RCM type [87,145,155,156]. These studies have used either one or a combination of various hydrological models to simulate hydrological responses under different climate change scenarios. Indeed, the best approach to understanding the hydrology of a river basin under various climate change scenarios is to employ various hydrological models under different scenarios [3]. However, Krysanova, Donnelly, Gelfan, Gerten, Arheimer, Hattermann and Kundzewicz [13] argued that in addition to using an ensemble of models, the best strategy is to use models once they have been evaluated and their performance is taken into account.

Unlike developed countries, a large number of developing nations do not have hydrological models of their own, largely because of fairly low technical capacity for simulating the models and lack of financial resources to compute and maintain the models [157]. This means that modellers and researchers in developing countries strive to choose an ideal and appropriate model from a variety of models developed in other regions.

The key question is what criteria modellers apply in selecting a hydrological model for different impact assessments. Addor and Melsen [158] realized that many researchers adopt a hydrological model due to the "legacy" associated with that model. The modellers are familiar and have practical experience, hence it becomes convenient to use such a model. Addor and Melsen [158] further reasoned that with time, the hydrological modeller gets a deeper insight into the model, hence among other things they can easily build it in a different setting and identify possible errors. However, using this approach will mean that the modeller will be limited to only using one model, despite the opportunities and potential gains from comparable models, which might even yield better results from the study objective perspective. Thus, model intercomparison where the strengths and weaknesses of each related model are evaluated is a necessary step to help with the choice of the right model. In addition, the relative easiness and flexibility of the model remain key factors in selecting a hydrological model. It is still common that a model is chosen due to that model and its codes being freely available to the public, the required quantity and quality of input data, the applicability in different regions, and the model's temporal and spatial scale [159].

Meanwhile, in developed countries, one country regardless of size may have numerous but similar hydrological models [160]. It can be argued that such diversity is due to different contexts in which the model is being generated and the physical features and geographical settings of the area under study. Still, even in some developed countries, despite having a number of models, most hydrologists are accustomed to using one hydrological model and seldomly pursue learning other models [160]. While it is known that there is no individual model that works for all scenarios [161], it is essential to apply different hydrological models to reduce the uncertainty associated with model structure and execution [162]. However, issues to do with environmental management including water resources, are complex in nature, hence scientists prefer working with only one suitable model [161].

Considering all these factors for hydrological model selection and having in mind that most countries do not have a model of their own, it is only rational that the preferred model particularly for climate change assessment should be capable of being transferred [163], such that the results can generalized to other regions and other time periods. Nevertheless, an appropriate hydrological model should serve its purpose, which includes understanding the catchment hydrological processes, testing the hypothesis, and also being used in the decision-making process [162,164]. Therefore, of ultimate importance would be that the model is precise such that the simulated parameters are closer to the observed values [164]. Although a model can be chosen based on one's preference, what is clearly important is that the model should achieve the objectives of the study, which are usually project dependent. Hence, one needs to understand if the model outputs are valuable to the project and whether the model can simulate the desired hydrological processes adequately. For example, a hydrological model projecting impact assessment on a river basin should be

capable to manage the continuous future changes in the basin, which are typically climate (temperature and precipitation) changes and land use/cover alterations.

In the southern Africa region, researchers on hydrological modelling have applied different models in areas of streamflow evaluations under different climate and land-use changes. The main reasons for the model choice range from a well-known history of the model's ability to achieve certain objectives, which is based on how the model was developed as well as evaluations from previous related studies, user-friendliness of the model and scalability of the model. Some recent studies on the hydrological models used in the southern African region and their reasons for adoption are summarized in Table 2.

**Table 2.** Summary of some of the recent studies in southern Africa applying hydrological models.

| Hydrological Model | Reason for Selecting the Model | Basin Applied | Reference |
|---|---|---|---|
| The Soil and Water Assessment Tool (SWAT) | Ability to perform long-term simulations, including climate change impact studies | Zambezi River basin | Ndhlovu and Woyessa [165] |
| The Water Evaluation and Planning (WEAP) | Comprehensiveness with respect to water resource management including the water supply, demand, and use. | Chongwe River Catchment, Zambia | Tena, et al. [166] |
| JAMS/J2000 model | Applicable for comparing different hydrological processes in different catchments, hence can be performed on medium and large-scale catchments | Verlorenvlei, Berg River, Eerste, Bot and Breede (Western Cape province, South Africa) | Watson, et al. [167] |
| mesoscale Hydrological Model (mHM) | Does not depend on the basin location and size (applicable on a basin scale of 4 to 530,000 km$^2$). Performed better in different catchments of different sizes and diverse climatic regions | Lake Malawi and Shire River Basins | Mtilatila, et al. [168] |
| The Pitman model | It has previously been utilized in the study area and continues to be used in South Africa for water resource planning | Eerste River catchment | Du Plessis and Kalima [156] |
| MIKE-SHE | Capability to model at a finer scale for small sub-basins | Upper Berg, Dwars, Du Toits, and Elands—all neighbouring catchments | Rebelo, et al. [169] |
| ACRU | The model has been widely used in the southern African region for studying climate and land-use changes | The uMngeni catchment | Kusangaya, et al. [170] |

*7.2. Commonly Used Hydrological Models for Climate Change Impact Assessment*

Across the world, the impact of climate change on streamflow variability continues to receive interest from a wide range of researchers [3,154,168,171,172]. Currently, varying hydrological models are used to simulate future changes in the hydrologic regime due to potential changing climate. Within sub-Saharan Africa where hydrometeorological data are usually scarce, some of the commonly applied models in literature for simulating streamflow include The Hydrologiska Byråns Vattenbalansavdelning (HBV) model [173], the Pitman model [156], the Variable Infiltration Capacity (VIC) [174], the SWAT [165,175], the HEC-HMS rainfall-runoff model [176,177], and most recently using the Artificial Neural Networks (ANN) as a tool for simulating runoff [178].

The HBV is a semi-distributed conceptual model, that requires daily rainfall, air temperature (in areas having snowfall), and monthly potential evapotranspiration data, in addition to catchment characteristics data such as digital elevation, land use, and vegetation cover [179,180]. It has been widely used for climate change and land use impact assessments. For example, the model was applied by Abdulahi, Abate, Harka and Husen [173] to investigate how future climate change would impact streamflow in the upper Awash sub-basin in Ethiopia. The CORDEX-Africa RCM temperature and precipitation data under the RCP4.5 and 8.5 scenarios were applied for the historical period 1996–2015 and streamflow was simulated for the 2021–2040 and 2041–2060 periods. The HBV model showed that the

streamflow in the Awash sub-basin would increase under both RCPs, largely in response to precipitation increase [173].

The Pitman model, which has been largely applied in southern Africa, is a monthly time step conceptual semi-distributed hydrological model, operating on a sub-basin whereby every sub-basin has its own data inputs [181]. Du Plessis and Kalima [156] applied the Pitman model to project the flow of the Eerste River (South Africa) due to climate change. Using the historical period from 1983 to 2018, Du Plessis and Kalima [156] simulated the temperature, precipitation, and flow for the 2022–2057 and 2058–2093 periods under the 4.5 and 8.5 RCPs. The study showed that as a result of climate change, evaporation will increase while precipitation will decrease resulting in an eventual decrease in streamflow.

A land-surface, semi-distributed VIC is one other model that has in recent times been used to model the impacts of climate change on the catchment hydrological regime, largely at a macro scale. The model, which requires rainfall, maximum and minimum air temperature, and wind speed is applied to simulate the effects of both land use and climate change in a river basin [180]. As it is a macro-scale model [182], it is inapplicable for small basins, hence it can project hydrological processes on wider catchment, regional, and even continental scales [183]. Roy, Valdés, Lyon, Demaria, Serrat-Capdevila, Gupta, Valdés-Pineda and Durcik [174] employed the VIC model in order to assess how the hydrological processes of the Mara River basin (East Africa) are impacted by climate change focusing on the near term. The model was able to predict the seasonal and annual trends in rainfall, evapotranspiration, and soil moisture.

The SWAT ecohydrological model is one of the most common, computationally efficient, and comprehensive river-basin models for tackling water resource problems in the long term [184]. The SWAT, a semi-distributed physically-based model, simulates different hydrologic processes and it can be applied in different basin scales, climatic regions, and land management practices. The model's main meteorological data requirements are daily precipitation, minimum and maximum temperature, wind speed, air humidity, and solar radiation while catchment characteristics such as slope, soils, and land use are also required for simulating the runoff [184]. Despite the popularity, validation of the model is a challenge particularly in ungauged basins since the model requires a large dataset. To overcome this, one common approach is regionalization, whereby data from a similar and gauged basin is used and transferred to an ungauged basin to calibrate and validate the model [185]. Further, the readily available global datasets for water resources, climate, and land use provide an opportunity for modelling in the SWAT platform.

The Hydrologic Engineering Center's Hydrologic Modeling System (HEC-HMS), is a semi-distributed conceptual and semi-physically based model for predicting flows at different spatial scales, including larger basins and small basins in natural and urban environments [186]. Lawin, Hounguè, N'Tcha M'Po, Hounguè, Attogouinon and Afouda [177] applied the HEC-HMS to model the impact of climate change on the flow of Ouémé River (Benin) between 1971 and 2050 by employing four global climate models. The HEC-HMS successfully projected the runoff due to climate change and at the same time took into consideration land-use changes. The study projected a significant decreasing trend and non-significant increase in streamflow based on the RCP 4.5 and 8.5 scenarios, respectively, hence proposed for the region to enhance its water infrastructure. The model was also recommended in catchments with limited soil data.

In recent years, one popular technique used in analyzing and estimating runoff under non-linear catchment processes is the use of artificial neural networks (ANN). ANNs use a mathematical simulation approach inspired by interconnected biological neurons [187] and are capable of representing complex and non-linear processes that connect a system's input and outputs [188]. The use of deep learning skills, therefore, can be highly used in place of the conventional conceptual or physically-based hydrological models, especially in regions with insufficient meteorological and topographical data and poor physical understanding of the basin characteristics [189,190]. Valeh, et al. [191] simulated the rainfall and runoff

in the Ammameh basin (Iran) under climate change using the ANN and SWAT models. Based on the error analysis, the study showed that the ANN model performed better than the SWAT including when determining extreme conditions such as floods and droughts. Hence, the ANN models can be assertively used in climate change studies, importantly in river basins with scanty hydrometeorological data, despite the need for high technical knowledge in programming.

It is of no surprise that regions such as sub-Saharan Africa largely employ semi-distributed hydrological models as they are not only physically based but are also relatively less demanding on input data. On that note, fully distributed models such as the MIKE SHE, despite being able to model the hydrologic processes in higher detail, are data-intensive and thus rarely applied in regions such as sub-Saharan Africa.

In summary, hydrological models continue to be used to address a wide range of water resources and environmental-related problems. However, the application of such models in regions such as sub-Saharan Africa is hampered by poor data availability especially at a microscale level, causing poor performance and high uncertainty [167]. At large, in most studies, the models are chosen based on data requirements and availability, and functionalities. For one to sufficiently describe the simulated phenomena such as future streamflow, the spatial and temporal extent of input data need to be adequate, otherwise the model's performance will be reduced. Owing to this, the use of gridded meteorological datasets offers a great opportunity in such basins where data is limited or poorly distributed. For instance, Onyutha, Turyahabwe and Kaweesa [175] showed that satellite-driven meteorological data can help in alleviating data paucity, which is a key challenge in applying physically-based hydrological models, particularly in developing countries. A review by Akoko, et al. [192] also indicated that with the use of remotely sensed data, the application of the SWAT model in ungauged basins in Africa was successful in analyzing runoff processes, simulating streamflow, and even calibration. Further, the use of Gravity Recovery and Climate Experiment (GRACE) satellite data remains a viable option for understanding and processing runoff characteristics [193–195]. Overall, despite the numerous models and techniques for simulating runoff under changing climate and catchment characteristics, each model will have its advantage and limitation, thus it is important to critically evaluate which model will help in achieving the study objective.

## 8. Uncertainty in Modelling the Hydrological Impacts of Climate Change

### 8.1. Uncertainties in Climate Models

The impacts of climate change are determined by the extent to which that change occurs. A thorough understanding of the historical, present, and future climate change impacts on water resources is hindered by the uncertainties associated with the multifaceted processes affecting the climate system [153]. Walker, et al. [196] defined uncertainty as "any deviation from complete deterministic knowledge of the relevant system". High uncertainty provides little confidence in the decisions made relating to both anthropogenic climate change and natural variability in climate. Therefore, minimizing uncertainty in future climate model simulations and their associated impacts on water resources is key to having reliable climate estimates and information, thereby ensuring better adaptation.

Uncertainty in modelling the hydrological processes can originate from various sources, which can happen in the past, present, and future estimates. Past and present uncertainties are largely due to two sources, which are data limitations and difficulties in determining the causes of complex processes in the hydrological system triggered by the interaction of biological, physical, and human systems [14]. This substantiates the need for having a broader understanding of the hydrologic system along with good quality and adequate data, with a view to building up a robust and reliable hydrological model.

Generally, from the climate system perspective, the two main sources of future uncertainty originate from firstly, the actions of human beings resulting in the undefined degree of future concentrations of emissions, and secondly, uncertainty over the capability and knowledgeability of humans to better model the climate [197]. Specifically, the different

steps applied to come up with the projections in regional climate models, collectively, also result in numerous uncertainties. These various sources of uncertainties from climate model projections can be categorized into four, namely [90]: (i) scenario uncertainty, i.e., uncertainty related to the emission or concentration scenario (ii) GCM uncertainty, i.e., uncertainty related to how the different global models respond to a particular emission scenario, (iii) RCM or downscaling uncertainty i.e., the uncertainty resulting from the use of several RCMs and several downscaling techniques from a specific GCM projection, (iv) uncertainty caused by the internal variability of the climate system.

Scenario uncertainty emanates from the unpredictability surrounding the emissions and concentrations of greenhouse gases in the atmosphere due to human activities [197]. Largely, scenario uncertainty increases tremendously with time, eventually becoming the most significant uncertainty when projecting climate change [99]. Human-induced factors such as population growth, industrialization, and related economic development will continue to increase the atmospheric burden in the next century [198]. Hence, it becomes difficult to confidently determine future concentrations as they depend on the conduct of humans. In addition to the above-mentioned varying sources of uncertainties, the overall uncertainty in climate projections is dependent on how these uncertainties integrate with different components of human behaviour, improvement in technology, and changes in socio-economic developments [149]. Still, despite such future uncertainties, the confidence in the climate model projections can be improved by a better understanding of the physical processes characterizing the models and also having a comparison of the previous forecasts against what was observed [99].

Understanding climate model uncertainty and increasing the level of confidence in the projections is key to adaptation planning. As already discussed, the unquantified anthropogenic activities and the resultant inherent scenario uncertainty remain a barrier to effective climate change adaptation planning. On this note, Motesharrei, et al. [199] argued that effective adaptation policies need to have a coupled earth/human system model by having bidirectional coupling, which would represent the real world's positive, negative, and delayed feedback. This would provide comprehensive knowledge of the modern non-linear world situation, which is important for science-based decision-making.

*8.2. Uncertainties in Hydrological Models*

Since hydrological models are a simplified version of the actual processes in the hydrological cycle, this simplification results in hydrological models becoming uncertain. Uncertainties in hydrological models originate from three key sources: parameter uncertainty, the structure of the model, and the input and observed or calibration data [162]. Considering that hydrological models involve the use of a wide range of input data, the model tends to be well parameterized, particularly as a result of the absence of such input data [92]. On the other hand, uncertainty due to model structure is a result of the inability to thoroughly simulate the real-world processes [200]. In essence, the performance of the model is largely dependent on the model structure [201].

Meanwhile, a better understanding of the hydrological system begins with the availability of comprehensive and accurate hydrological data. McMillan, et al. [202] summarized the uncertainties in the hydrological data into five groups, namely: measurement or point uncertainty e.g., rainfall depth whose measurement occurs at a point; uncertainties from data derived from a proxy measurement, such as streamflow derived from a river stage measurement, uncertainty from data interpolated in space and time, scaling uncertainty whereby data measured at one usually small scale is used for a process at another larger scale, and uncertainties in data management, for example, due to human or computing errors.

From the above reviews, it is shown that hydrological simulations under climate change are limited due to uncertainties from both hydrological and climate models. As there are varying sources of uncertainties, it is likely that some types of uncertainties have a larger influence on the final model output than others. For instance, Joseph, Ghosh, Pathak and Sahai [92], while assessing the hydrologic impact of climate change, compared the

contribution of parameter uncertainty and climate model uncertainty in the hydrological simulations. The study found that hydrological parameter uncertainty was less and nearly negligible compared to climate model uncertainty, hence suggesting the need to prioritize minimizing the climate model uncertainty before adopting the findings. Indeed, reducing these uncertainties provides high confidence to policymakers to use the projections for managing water resources and adapting to climate change. Dobler, et al. [203] also identified that choices of the GCM and RCM were the principal sources of uncertainty while the uncertainty from hydrological model parameters was negligible. Therefore, to reduce these uncertainties, it is recognized that future climate change projections cannot be based on a single model. Therefore, in order to develop an acceptable model, multiple simulations from multi-model ensembles are used [154,204]. This is because the various structures in different models pick up a wider range of catchment responses, which are subsequently captured in the combined model [201].

## 9. Conclusions

Despite recent improvements in the horizontal resolution in global climate models, on their own, these models may be unable to effectively simulate the hydrological impact of climate change at a river basin scale. Thus, climate models are integrated with hydrological models in order to adequately describe the responses of the basin due to both climatic change and human influences. Under changing physical and natural environments, hydrological models continue to be valuable tools for developing different scenarios for managing water resources, thus, choosing the right hydrological model is crucial.

The impact of climate change on water resources and mankind, in general, will be determined by how the earth system responds to variations in radiative forcing but also on the society's feedback and adaptation to changes in technology, demographics, socioeconomic development, and policies. High uncertainties in the future radiative forcing have facilitated the continued development and use of the latest and improved emission scenarios, which are used to assess the prospective impacts of different policies and response plans. Thus far, RCPs and SSPs present a great opportunity for advancing climate research and they offer a potential framework for mitigating emissions and analyzing the impact of climate change. For an individual study where climate models are coupled with hydrologic models, it is essential to integrate the RCPs and SSPs for effective assessment of climate risk, adaptation, and mitigation.

From a water resource modelling perspective, the initial uncertainties stem from differences in the spatial and temporal coarser resolution of the climate model compared to the small-scale and finer resolution of the hydrological model. Meanwhile, information at a smaller scale is the most preferred for better planning at a river-basin scale. Reducing uncertainties in modelling the hydrological impacts of changing climate largely involves the advancement in data availability and improvement in both the climate and hydrological models.

As reviewed in this article, model input data can be improved by incorporating remotely sensed and other global datasets while model improvements would entail having a better understanding of the model's physical processes, assumptions, and limitations in the applicability of the model. This is critical for developing countries to better adapt to the ever-changing environment. For example, within the southern Africa region, many recent articles have incorporated different downscaled RCMs to improve the modelling results. Further, the best-performing RCMs have been indicated in several studies. Nevertheless, more research on the performance of downscaled RCMs at a basin level needs to be intensified because it is at this scale where most decisions on water resource management are carried out.

**Author Contributions:** V.D.B. (conceptualization, planned the scope and structure, writing—original draft preparation), R.B.D. (conceptualization, funding acquisition, resources, supervision, writing—review and editing), S.K.S. (conceptualization, supervision, writing—review and editing), T.K. (conceptualization,

resources, supervision, writing—review and editing). All authors have read and agreed to the published version of the manuscript.

**Funding:** The financial assistance of the South Africa National Research Foundation (NRF) is hereby acknowledged. Opinions expressed and conclusions arrived at, are those of the authors and are not necessarily to be attributed to the NRF. The research is on the Rietspruit sub-basin under the grant BRICS multilateral R&D project (BRICS2017-144), the NRF UID number 116021 and the Durban University of Technology UCDG Water Research Focus Area grant. The research was also financially supported by the University of the Western Cape under the Research Incentive Funds.

**Data Availability Statement:** The information on data presented in this study are available on request from the corresponding author.

**Acknowledgments:** The BRICS multilateral R&D project (BRICS2017-144) Team is sincerely acknowledged. Raghavan Srinivasan and the SWAT Team at TAMU Agrilife Research, USA, and Ann Van Griensven and her Research Team, Department of Hydrology and Hydraulic Engineering, The Vrije University, Brussel, Belgium, are all sincerely acknowledged. The Durban University of Technology is sincerely acknowledged for hosting the grant. University of Limpopo and South Africa's Agricultural Research Council-Natural Resources & Engineering are most sincerely acknowledged as co-investigators in this NRF-BRICS research project.

**Conflicts of Interest:** The authors declare no conflict of interest.

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
