# Peer review of "Hydrological Modelling and Climate Adaptation under Changing Climate: A Review with a Focus in Sub-Saharan Africa"

_water, doi:10.3390/w14244031_

Round 1
Reviewer 1 Report
General comments
Thank you for the opportunity to review this article entitled “Hydrological Modelling and Climate Adaptation under Changing Climate: A Review” by Banda et al.
Overall this manuscript is very shallow and, while I understand that this is intended to be a review article, there is no real summary or collection of information. No quantitative data are given, and provided figures and table are simply reposting of previous publication with no additional work.
My main concern is the complete lack of methodology concerning the review. In fact, there is no materials and methods section and hence the selection of article used to produce this review is completely arbitrary. The review also ignores certain field such as global hydrology despite their relevance.
Because the authors do not focus solely on one aspect of climate change (i.e., flood, drought…) the manuscript remains vague, does not go into details, and provides only simple “facts” (backup with limited and arbitrary citation). As a result, it is extremely shallow as demonstrated by the complete lack of any numbers. I was expected to find some range regarding the impact of climate change (people affected, damage…) given various adaptation and mitigation scenarios. Focusing on “hydrological modelling” and climate change and adaptation, I expect the citations to be in the order of thousands, not a hundred as showed here. Hence my advice to the authors to refocus the article on a particular aspect of climate change (flood, drought, water scarcity…)
The delineation of sections in the article is rather arbitrary and the fact that this article focuses on developing countries should absolutely be reflected in the title!
As a result, I recommend to reject this article. The authors need a lot of time to produce a focused scientific article.
Specific comments
Abstract
L20-21: Some items are redundant
Introduction
L41: why do you mentioned climate variability here?
L43: what exactly is a “concern”? do you mean that there are numerous studies?
L44: detected would be more appropriate. Possibly mentioned the field of detection and attribution which focuses exactly on that problem
L48-49: the last sentence is redundant and should be removed
L50: In particular to what? This transition is awkward, revise the sentence.
L61: long term -> long periods possibly more appropriate
L63: why only one GCM? Most studies use an ensemble (multiple) of GCMs.
L64: same remark for emission scenario; typically studies employ several of them
L67: precise the order of magnitude of short/long time scale
L72-79: the sections you announced here are not clearly visible in the, manuscript afterwards
Climate change mitigation adaptation and its impacts in developing nations
L84: you should provide some concrete examples
L85: i.e is not necessary in this context
L90-92: Are you sure this is accurate? This is the opinion of one paper which is quite old now. Many government of developed country have established adaptation strategies to climate change.
L99: Which review!? How articles were selected, how many, from which country for what topics? There is no method!
L100-101: what do you mean?
L107: what do you mean by trusty? Accurate? Reliable?
3. Nexus between Climate Change, Land Use, and Hydrological Processes
L129-130: It is important to precise that assumptions regarding socio-economic largely outweigh to climate change assumptions for water risk in general (i.e., flood, drought, water scarcity)(see https://hess.copernicus.org/articles/22/789/2018/)
L144-146: this is already the case. All flood risk analysis use population data (an example here: https://www.nature.com/articles/nclimate1911)
L150: only quantity important? At what temporal resolution and what about timing…
L154-155: This is a standard procedure is all “detection and attribution” studies which need to disentangled the effects of climate change, natural variability, and anthropogenic activities. See the ISIMIP simulation protocols (https://www.isimip.org/)
L157-161: What are you speaking about? Both reference 43 and 44 rely on the SWAT model. The statistical technique deployed for identifying a (significant) change depend on the type of data available and on the experimental framework.
L159: SWAT not defined
L165-168: Again this is only one reference. Hydrological model differ greatly among each other (see this https://gmd.copernicus.org/articles/14/3843/2021/ )
4. Climate Change Modelling
L187: this is the resolution in the atmosphere! More relevant is the lon*lat resolution typically in the order of 50km*50km
L195-196: You focus purely on downscaling but what about bias-correction?
L196-199: why do you ignore the rich literature of global studies? One example here (https://www.science.org/doi/10.1126/science.aba3996)
5. Assessing Future Climate Change: Climate Change Scenarios
What is the purpose of the section? This information was presented in multiple previous publications, is widely available and you add absolutely nothing to it.
L341-348: why do you not mentioned ensemble?
L393: hence ensembles are needed
6. Synopsis of Applied GCMs and RCMs in Southern Africa
This is one of the more interesting section but it completely lacks quantitative information.
L422-424: bias correction (https://www.nature.com/articles/s41598-021-82715-1) is very important but you only devoted 2 lines to it!
7. Hydrological Models under Changing Climate: Role of Hydrological Models In Climate Change Studies
Since you did not disclose how article were selected this section leaves an impression of subjectivity. A good example is L449: only one author noted this point is that relevant in the grand scheme? What about the process that re or are not considered? Despite SWAT being know to not simulate rice paddy process appropriately, the model can “accurately” reproduce daily streamflow (https://ijabe.org/index.php/ijabe/article/view/7147/0). It means that tweaking or “calibrating” some parameters may hide some fundamentally broken process. Problem arise when the same model is used to predict future condition (new climate or new land use) as the model is perceived to be accurate but future projections may be completely of. Also of interest: https://link.springer.com/article/10.1007/s10584-016-1829-4
L475-482: seems reasonable. WWhat about other reasons: i) open source status (For example, SWAT code can be downloaded and fairy easily modified if necessary), ii) access to documentation and community/help is tremendous, iii) ease of use/integration to GIS/graphical interface (in contrast with command line interface only models). I believe that these are partial reasons why the SWAT model is so prevalent worldwide. This section should be strengthen to be more “scientific”, as mentioned above this is but one publication show a table for key countries with the number of time each model was applied for example…
I note that since you stick with no specific topic (flood, drought) it sound impossible to find a best rounded model.
L493-496: Develop countries also rely heavily on SWAT and models not develop in their country that this is not specific to developing countries.
L501-507: this is so basic; I do not need to read any reference to tell you that.
L526: ANN is strictly speaking not a hydrological model.
Generally, how can I be convinced that those are the most represented models when you reported nothing about the methodology used to screened article and conduct the review.
L562-563: calibration process can be automated. If no data available, then no luck but this is true with other models as well.
L596-612: what about upcoming satellite imagery (GRACE)
8. Uncertainty in Modelling the Hydrological Impacts of Climate Change
Too generic and no real synthesis of available knowledge.
L655-662: this is already a reality: https://gmd.copernicus.org/articles/13/4713/2020/
9. Conclusions
Overall this section brings absolutely no new knowledge and is very shallow.
L698: where did you get this resolution? From this article (https://www.sciencedirect.com/science/article/pii/S1674927821001477) this is the spatial resolution in the atmosphere. More relevant is the horizontal resolution which varies by model but is generally in the order of 50km by 50km.
L698-700: this need revising
L703: Only beginning? they have been valuable tools for quite a while now.
L705-708: this is very generic, it is appropriate in the conclusion
L709-718: What is new? This is very generic
L719: is that all? What about processes?
L726: mentioned global dataset however the entire literature devoted to global hydrology is utterly ignored!
Figures
Fig.1: what do the numbers in bracket represent? Do you have the permission to reproduce such figure, better to adapt the figure.
Table 1: this table is so small, why these article in particular there are thousands of article published every year that leverage these models!

Author Response
We would like to thank the reviewer for the comments, suggestions, and their time in critiquing this paper, which we believe has strengthened the paper. All the revisions and responses to the comments and suggestions by the reviewer have been summarized in the document attached.

Reviewer 2 Report
This review paper revealed the application and current limitations of climate models and hydrologic models under climate change. The paper first introduced current conditions of climate change and corresponding mitigation and adaptation worldwide and then pointed out the strategic significance of numerical modeling on adaptations. Focusing on Southern African regions, what the standards of right hydrologic models are and what the commonly used hydrologic models are represented. After this, the authors proposed the sources of uncertainties from hydrologic and climate models and the possible methods for reducing uncertainties to improve the adaptations.
I would like to point out some aspects of the manuscript that I would like the authors to clarify. Also, I would like to make some questions and suggestions about some points I could identify in the manuscript. Suggest authors improve the representations to enhance the logical relationships between paragraphs and sentences.
These are listed comments:
11. Line 91 – Line 94: “Adaptation to climate change has different…. During times of hazards”. It would be better to show several examples following this, showing how adaptation is applied and focused in developing nations.
22. Line 94 – Line 95: “Hence, vigilant …. of countries”. The causality between this sentence and the previous sentence is weak. Please modify the representation.
33. Line 81 – Line 82: “Sub-Saharan Africa is …and sectors” and figure 1 talks about the conditions in Africa. But the title of the session is for developing nations. Suggest changing the session title or providing more content about developing nations in other regions and providing a richer discussion of the commonness of developing nations on adaptation measures.
44. Line 165 – Line 168: “for example, Guan, et al.…streamflow changes”. This example looks not very appropriate logically. Why is this the example of using a mix of approaches due to limitations in each individual approach? Please clarify this.
55. Line 164: Also, before talking about “limitations”, it would be better to mention what the limitations are.
66. Line 193: “Therefore, it is important…” There is weak causality between this sentence and the previous one. Suggest changing the representation.
77. Line 324: “Soil Water Assessment Tool (SWAT)”. Here is not the first-time mentioned SWAT. The full name of SWAT should be explained when it was mentioned for the first time (Line 159)
88. Line 519- 521: “Currently, varying … felt simultaneously”. Is there any reference literature on this conclusion that the impacts of potential changing climate and land use are felt simultaneously? This sentence itself is also confusing. What is the reason for putting it here?
99. Line 545: “Variable Infiltration Capacity (VIC)”. Not necessary to show the full name here since it was explained in Line 524
110. Line 628 – 630: “Hence, good … reliable one”. Weak causality. The previous content will not get this result. Please improve the representation.
111. Suggest adding one paragraph at the end of the introduction session to simply announce in advance what the readers can see in the following sessions and further clarify the logic of the structure of the paper, for example, what is the internal relationship among sessions.
Author Response

(The authors gave the same response as above.)

Round 2
Reviewer 1 Report
I appreciate the effort of the authors in addressing most of my specific comments and suggestions. However, as I made clear in my previous review, the critical flaw in this manuscript is the absence of a material and method section and this point was not addressed by the authors. This is simply not acceptable and many articles provide guidelines on how to conduct systematic literature review.
- here is one: https://www.ncbi.nlm.nih.gov/pmc/articles/PMC6974768/
Alternatively, there are also numerous existing review articles dealing with climate change which methods could be adapted and implemented into this manuscript:
- https://link.springer.com/article/10.1007/s11356-022-19718-6
- https://www.tandfonline.com/doi/full/10.1080/23789689.2019.1593003
- This article is particularly relevant as it target Thailand, a developing country: https://iopscience.iop.org/article/10.1088/1748-9326/abce80/meta
The second critical problem is the lack of focus of this manuscript. It is challenging enough to publish an article focusing on a single country (see above) but targeting all developing country or Africa (which I strongly insist MUST be reflected in the title) sound borderline impossible given the sheer amount of articles that must be read and integrated into the review should the authors follow a sound review method.
Again, the authors must adopt a systematic literature review for targeted countries/regions, otherwise the selection of presented article simply depend on the mood of the authors (where the article mentioned above cited, if not why) and demonstrate nothing.
Author Response
The Authors' notes to the Reviewer are provided in the attached file.

Reviewer 2 Report
The questions were well answered. Some minor issues with the format:
Please check the font size of the manuscripts and keep them consistent. For example, L130-L142, L248, L527, and so on.
Author Response
We have edited the formatting errors in the document.
We thank the Reviewer for support and guidance.
Kind regards
Round 3
Reviewer 1 Report
Similar to my previous review, this manuscript is just not focused at all. The authors want to present all aspects of climate change and hydrology following a very subjective method to arbitrary select article. So if different articles were selected would the content of this manuscript change? How can this approach be scientific? As a result, it is extremely shallow and is simply a collection of obvious facts.
The authors must ask this question: why would someone read this article instead of a focus review article on “flood under climate change” or “drought under climate change”, or “water scarcity under climate change”… For example if I want to know everything about bias correction, why should I read this article and not a “specialized article” like this one; “An empirical evaluation of bias correction methods for palaeoclimate simulations”. The unique perspective of Sub-Saharan Africa is a good idea but so far the discussion is not tailor to this area, again because of the lack of a focus topic. Look at the ssp section, you could write the exact same paragraph for any regions of the world.
---
Abstract: you claim that you discuss “choosing the appropriate global climate models, 24 downscaling techniques, emission scenarios…” this is not the case. This manuscript is just a shallow list and collection of facts and we have absolutely no idea how these findings demonstrated for an isolated study or at larger, or global scale could transfer to the sub-saharan region. Again this is mainly because you did not select a focus research topic. For analysisng flood, some models must perform better that other and some climate, bias technique would be better appropriate (extreme matter but low and average condition are less relevant for example).
L26-27: does not bring new info and what conventional means? Remove
L34-35: this is already implemented in many models
L54: population is too restrictive. Socio-economic condition is more appropriate
L65: what exactly do you mean by water resources? Are drought, flood included?
L76-79: you need to specify somewhere the target region(s)
L88-91: what did you used exacly. This is too vague. (“sub-Saharan Africa” OR “Southern Africa”) AND (“climate change” OR “climate model” OR “…”). I do not buy the “narrative literature” approach. This is a scientific journal at the end of the day and a scientific methodology MUST be applied and followed.
L93: yes so many because this study is not focused and it is not clear what is and is not analyzed. Again are flood droughts, extreme events included? Water resources are also critical for agriculture are those paper included… At least you included this section but there is no quantitative numbers nor analysis given hence it is not that convincing. At the very least report how many papers were identified through the first query for each search index. Then let reader know how many you decided to retain (about 200 it seems).
L104-107: this must be mentioned in the abstract as well
L153: water scarce period
L155: what about possible negative environmental repercussion of these measures? For example, dams are known to cause many problems.
L161: and change in socio-economic
L168-169: how can you say that? What about maladaptation and limit to adaptation?
L210: what about timing?
L270: and consumption, withdrawal, pollution…
L316: what do you mean by hydrologic variability at a regional scale? Global models are applied at lower scale such as continental and regional…
L353: what does downscaling the hydrological cycle mean?
L648: why “appropriate”” when many of these models are open source
L630: how accessible the source code of these models must matter as well.
L910-916: such findings are region dependent. Are those relevant for this review within the context of Africa?
Author Response
We would like to thank the Reviewer for the additional comments and questions raised in the manuscript, which are crucial to improving this paper.
Most of the new items raised have been attended to.
As earlier mentioned, the paper aimed to provide a comprehensive analysis and understanding of the current knowledge on a topic, hence the approach taken in this manuscript.